# Is Citicoline Effective in Preventing and Slowing Down Dementia?—A Systematic Review and a Meta-Analysis

**DOI:** 10.3390/nu15020386

**Published:** 2023-01-12

**Authors:** Maria Bonvicini, Silvia Travaglini, Diana Lelli, Raffaele Antonelli Incalzi, Claudio Pedone

**Affiliations:** 1Training Programme in Geriatrics, Department of Medicine and Surgery, Università Campus Bio-Medico di Roma, Via Alvaro del Portillo 21, 00128 Roma, Italy; 2Training Programme in Internal Medicine, Department of Medicine and Surgery, Università Campus Bio-Medico di Roma, Via Alvaro del Portillo 21, 00128 Roma, Italy; 3Research Unit of Geriatrics, Department of Medicine and Surgery, Università Campus Bio-Medico di Roma, Via Alvaro del Portillo 21, 00128 Roma, Italy; 4Operative Research Unit of Geriatrics, Fondazione Policlinico Campus Bio-Medico, Via Alvaro del Portillo 200, 00128 Roma, Italy; 5Operative Research Unit of Internal Medicine, Fondazione Policlinico Campus Bio-Medico, Via Alvaro del Portillo 200, 00128 Roma, Italy

**Keywords:** mild cognitive impairment, Alzheimer’s disease, vascular dementia, citicoline

## Abstract

Background: Cognitive impairment is a staggering personal and societal burden; accordingly, there is a strong interest in potential strategies for its prevention and treatment. Nutritional supplements have been extensively investigated, and citicoline seems to be a promising agent; its role in clinical practice, however, has not been established. We systematically reviewed studies on the effect of citicoline on cognitive performance. Methods: We searched the PubMed and Cochrane Library databases for articles published between 2010 and 2022. Relevant information was extracted and presented following the PRISMA recommendations. Data were pooled using the inverse-variance method with random effects models. Results: We selected seven studies including patients with mild cognitive impairment, Alzheimer’s disease or post-stroke dementia. All the studies showed a positive effect of citicoline on cognitive functions. Six studies could be included in the meta-analysis. Overall, citicoline improved cognitive status, with pooled standardized mean differences ranging from 0.56 (95% CI: 0.37–0.75) to 1.57 (95% CI: 0.77–2.37) in different sensitivity analyses. The overall quality of the studies was poor. Discussion: Available data indicate that citicoline has positive effects on cognitive function. The general quality of the studies, however, is poor with significant risk of bias in favor of the intervention. Other: PubMed and the Cochrane Library.

## 1. Introduction

Mild cognitive impairment (MCI) is defined as an objective decline in cognitive function, reported by a patient or relative, that has no or minimal impact on instrumental activities of daily living [1]. Some people with MCI will progressively develop dementia while others remain stable or recover full functioning [2]; it is estimated that about 12% to 36% of people aged 65 and over have MCI [3] and, as the population of older adults increases, the prevalence of MCI will gradually increase [4]. People with MCI are at greater risk of dementia than the general population; the mean annual conversion rate of MCI to dementia is approximately 10% [5], which is far higher than the annual incidence (1–2%) in the general population. According to the existing literature, MCI progresses mostly to Alzheimer’s disease and, less frequently, to vascular dementia [6].

Alzheimer’s disease (AD) is the most common cause of dementia, accounting for an estimated 60% to 80% of cases (50 million worldwide) [7], and its prevalence is increasing worldwide. AD is characterized by intraneuronal fibrillary tangles and extracellular deposit of amyloid plaques (Aβ) coupled with reactive microgliosis and loss of neurons and synapses in the cortex [8]. Deposits of Aβ can lead to cortical dysfunctions resulting in several cognitive impairments such as memory and intellectual disabilities, causing a disability in activities of daily living [9] and interfering with quality of life.

Another common cause of dementia is vascular dementia (VaD), accounting for about 20% of all cases of dementia [10]. It is a neurocognitive disorder with significant cognitive impairment that is directly related to vascular injury to the brain; it has several potential contributing factors such as ischemic stroke [11]. Almost half of stroke survivors have cognitive impairment [12]; cognitive decline after stroke is even more common than stroke recurrence, and stroke patients with cognitive impairment but no dementia have an increased risk of developing VaD and other dementias at 5 years [13].

MCI and dementia can also occur in patients with Parkinson’s disease (PD); the prevalence of PD-MCI ranges from 20% to 40% depending on the population studied, and the incidence of PD-dementia increases with the duration of disease, with estimates as high as 30% at 5 years and over 80% at 20 years from diagnosis [14,15]. Cognitive decline is usually slow and insidious, but may be rapid in some cases [16].

MCI and dementia pose significant challenges to patients, caregivers and healthcare systems due to their rising prevalence and socioeconomic burden, with the numbers of people living with dementia worldwide projected to triple to 152 million cases by 2050 [17]; furthermore, MCI [18] and dementia [19] seriously affect the quality of life and well-being of older adults. In the face of this staggering impact on personal and societal health of these conditions, the therapeutic options are somewhat limited [11,20,21], and there is a strong interest in potential strategies for prevention and treatment interventions.

It has been suggested that dementia may be prevented or slowed down by improving nutritional status. For example, Dominguez et al. [22] documented that adherence to the Mediterranean dietary pattern and Dietary Approach to Stop Hypertension (DASH) is associated with slower rates of cognitive decline and significant reduction in incidence of AD. In addition, Martinez [23] suggests in a systematic review that supplementation of B complex vitamins, especially folic acid, may have a positive effect on delaying and preventing the risk of cognitive decline. Given their safety, nutritional supplements have been extensively investigated as an option for treating cognitive disorders [24,25,26,27,28], especially in the initial stages of the disease to prevent the progression to dementia [29,30]. A consensus has emerged that intervention strategies must be initiated as early as possible, even before any significant symptoms begin to appear [31,32].

One of the most promising nutritional supplements is citicoline, the pharmaceutical form of the endogenous compound cytidine-5′-diphosphate (CDP) choline. It increases the intrasynaptic concentration of acetylcholine and promotes phospholipid synthesis and neuronal repair, as shown in different studies in animals [33]. Moreover, it can inhibit apoptosis associated with cerebral ischemia and several models of neurodegeneration, as demonstrated in studies in vitro or in animals [34]. It has neuroprotective properties, as shown in studies performed in vitro or in animal models, that have been shown in different patterns of cognitive impairment, Parkinson’s disease or dementia (Alzheimer’s disease or vascular dementia), and that are much more evident when the treatment is administered for an extended period of time [35]. Citicoline seems to be effective also in provoking cognitive functions in normal healthy people as shown by Al-kuraishy et al. [36]; it improved human psychomotor vigilance, arousal and visual working memory with significant amelioration of oxidative stress after two weeks of being administered in healthy volunteers (age range 21–22 years).

The current evidence on this supplement has been obtained from studies that for the most part have a small sample size, and has been summarized in previous reviews that have focused on Alzheimer’s disease [37], or on MCI associated with Parkinson’s disease [38] or on a wide range of neurological conditions [39]. The results are heterogeneous; furthermore, there are no reviews that compare and analyze at the same time the efficacy of citicoline in MCI and in different types of dementia.

The aim of this systematic review and meta-analysis is to evaluate whether citicoline, compared with a placebo or standard treatment, is effective in preventing dementia in people with MCI or in improving cognitive functions in people with dementia.

## 2. Materials and Methods

### 2.1. Search Strategy and Selection Process

The study was conducted according to the PRISMA [40] statement recommendations. We searched PubMed and the Cochrane Library for relevant articles in October 2022. The search strategy was based on the following keywords: “citicoline”, “cytidine diphosphate choline”, “choline”, “cognitive impairment”, “cognitive decline”, “cognitive dysfunction”, “Alzheimer’s disease”, and “vascular dementia” (Appendix A). We restricted the search to clinical trials using “Clinical Study”, “Clinical Trial”, “Randomized Controlled Trial” as a filter, and limited the search to articles published in English between 2010 and September 2022.

This search strategy retrieved 512 articles, of which 48 were duplicates. The titles and abstracts of the 464 remaining articles were independently reviewed by two of the authors (MB, ST) to verify inclusion and exclusion criteria. Inclusion criteria were based on the PICO framework and are reported in Table 1. The full text of 34 records was then carefully reviewed and assessed for eligibility; the bibliography of each article was examined. Finally, 7 articles that met the inclusion criteria were included in this systematic review. The detailed process is depicted in Figure 1.

### 2.2. Inclusion Criteria

The inclusion criteria for this systematic review were (a) clinical trials or randomized controlled trials; (b) presence of control group; (c) quantitative outcomes; (d) publication date (from 2010 to September 2022); (e) studies performed on humans; and (f) studies in the English language.

### 2.3. Data Collection and Analysis

From each of the included studies, we extracted information on methodological characteristics (author, publication date, study design, number of participants), interventions (type of intervention including dose, frequency, and duration; concomitant treatments, type of control used) and participant characteristics (baseline cognitive status or disease, age, and gender)—see Table 2. The outcome domains considered were cognitive functions. In all included studies, the results matched the cognitive domains stated in the objectives.

The Cochrane Collaboration’s tool was used to assess risk of bias (adapted from Higgins and Altman) [42] and the overall risk of bias—see Table 3. The risk bias tool covers six domains of bias: selection bias, performance bias, detection bias, attrition bias, reporting bias and competing risk. Two reviewers have assessed the risk of bias in each study and worked independently; disagreements were resolved by consensus.

The results from the included studies were pooled using the inverse-variance method with random effects and multilevel models considering the correlation of results coming from the same study. Most studies tested and reported the within-group change in outcome measures. When possible, we used the reported *p*-values to calculate standard deviation of the within-group difference. When the standard errors or *p*-values were not provided, we used the following formula to derive the standard deviation [43]:$$SD_{change}=\sqrt{SD_{baseline}^{2}+SD_{final}^{2}-{\left(2\times Corr\times SD_{baseline}\times SD_{final}\right)}_{}{}^{}}$$

The correlation coefficients were calculated hypothesizing both a high (0.9) and a low (0.6) correlation between observations.

For the meta-analysis including all the studies, we used the standardized mean difference of the outcome measure to account for the different measures used. Sensitivity analyses were performed by analyzing studies with the same design (randomized vs. observational) and studies using the same outcome measures separately. We repeated the analyses under the two different scenarios used to calculate the standard deviations. Funnel plots (Appendix A) were used to visually assess bias.

**Table 2 nutrients-15-00386-t002:** Characteristics of the studies included.

Authors, Year	Study Design	Participants and Mean Age	Target Disease	Outcome Measures	Results
Castagna et al., 2016 [44]	Case–control study	174 patients (mean age: 81.3 ± 4.5 years) divided into two groups. CASES (rivastigmine + citicoline 1000 mg/day orally): 92 patients (62 affected with AD, 30 with MD; 29% men). CONTROLS (rivastigmine): 82 patients (53 affected with AD, 29 with MD, 28% men).	Alzheimer’s disease and mixed dementia	Primary outcomes: effects of combined administration versus rivastigmine given alone on cognitive functions. Cognitive functions were assessed by MMSEc score which had been administered at baseline, 3 and 9 months. Secondary outcomes: possible side effects of combination therapy versus rivastigmine alone.	MMSEc in AD patients (cases): T0 15.68, DS 3.03, T1 16.79, DS 2.84, T2 16.93, DS 3. MMSEc in AD patients (controls): T0 15.32, DS 3.55, T1 14.81, DS 3.58, T2 13.97, DS 3.56. MMSEc in MD patients (cases): T0 16.04, DS 3.13, T1 16.41, DS 3.26, T2 16.62, DS 3.55. MMSEc in MD patients (controls): T0 14.79, DS 2.75, T1 14.33, DS 2.96, T2 13.2, DS 2.62.
Castagna et al., 2021 [45]	Multicentric, case–control study	170 patients (mean age 76.8 ± 4.93 years, 34.11% men) divided into two groups. CASES (citicoline 1000 mg/day orally + memantine + AChEI): 81 patients (47.65%). CONTROLS (memantine + AChEI): 89 patients (52.35%).	Alzheimer’s disease	Primary outcomes: to assess whether a combined treatment of citicoline, memantine, and AChEI slows cognitive impairment. Cognitive functions were assessed by MMSE score which had been administered at baseline, 6 and 12 months. Secondary outcomes: to assess (a) safety and adverse drug reactions; (b) the possible interactions of citicoline, memantine, and AChEIs with other drugs.	MMSE score in the treated group had a statistically significant increasing trend between T0 and T2: 14.88 (DS 2.95) at T0, 14.95 (DS 2.63) at T1, 15.09 (DS 3) at T2. MMSE score in the control group showed a statistically significant decrease trend: 14.37 (DS 2.63) at T0, 14.19 (DS 2.81) at T1, 14.03 (DS 2.92) at T2.
Alvarez-Sabin et al., 2016 [46]	Open label, randomized, parallel study	163 patients (83 women, 50.9%), mean age 67.5 ± 10.7 years, divided into two groups. CASES (citicoline 1 g/day orally): 86 patients (52.8%) CONTROLS: 77 patients (47.2%).	Post-stroke (first ischemic stroke)	Primary outcomes: cognitive status and quality of life.	Citicoline group showed a significant improvement in cognitive status during follow up (GCI 43.6% at 1 month, 32.5% at 6 months, 29% at 1 year, 27.9% at 2 years). The untreated group did not show significant changes (42.3% at 1 month, 41% at 6 months, 38% at 1 year, 39% at 2 years).
Cotroneo et al., 2013 [47]	Open label, multicentric study	349 patients were divided into two groups. CASES (Citicoline 500 mg bid orally): 265 patients (122 men and 143 women), mean age 79.9 ± 7.8 years. CONTROLS (no treatment): 84 patients (36 men and 48 women), mean age 78.9 ± 7.01.	Mild vascular cognitive impairment	Primary outcome: the effect of citicoline on cognitive functions. Cognitive functions were assessed by MMSE score which had been administered at baseline, 3 and 9 months.	The MMSE score in the treated group remained essentially unchanged over time (22.4, DS 4 at T0; 22.7, DS 4 at T1; 22.9, DS 4 at T2). The control group showed a decline in MMSE score over the 9 months (21.5, DS 6.9 at T0; 20.4, DS 6.6 at T1 and 19.6, DS 6.3 at T2).
Gareri et al., 2016 [48]	Multicentric, case–control study	448 patients (39.03% men, 60.97% women), mean age 80.03 ± 6.77 years, divided into two groups. CASES (AChEI—donepezil or rivastigmine or galantamine + citicoline 1000 mg/day orally): 251 patients. CONTROLS (AChEI): 197 patents.	Alzheimer’s disease	Primary outcomes: effects of combined administration versus AChEIs given alone on cognitive functions. Cognitive functions were assessed by MMSE score; it was administered at baseline (T0), after 3 (T1), and 9 months (T2). Secondary outcomes: possible side effects or adverse events of combination therapy versus AChEIs alone.	MMSE in the treated group: 16.88 (DS 3.38) at T0, 17.62 (DS 3.64) at T1, 17.89 (DS 3.54) at T2. MMSE in the control group: 16.41 (DS 2.97) at T0, 15.99 (3.16) at T1, 15.41 (DS 3.16) at T2. They compared MMSE scores in the treatment group in order to assess the possible differences among the three different AChEIs. MMSE score in donepezil group (144 patients): 17.15 (DS 3.83) at T0, 18.15 (DS 4.21) at T1, 18.49 (3.98) at T2. MMSE score in rivastigmine group (105 patients): 16.54 (DS 2.64) at T0, 16.89 (DS 2.53) at T1, 17.07 (DS 2.66) at T2.
Castagna et al., 2021 [49]	Multicentric, case–control study	104 patients (76.04 ± 4.97 years; males 27.88%) divided into two groups. CASES (citicoline 1000 mg/day orally + memantine + rivastigmine): 41 patients. CONTROLS (memantine + rivastigmine): 63 patients.	Alzheimer’s disease	Primary outcome: to assess whether or not triple therapy (citicoline, memantine and rivastigmine) slow cognitive impairment progression. Cognitive functions were assessed by MMSE score; it was administered at baseline (T0), 6 months (T1) and 12 months (T2). Secondary outcomes: safety and possible side effects.	MMSE case group: 13.63, DS 2.46 at T0; 14.17, DS 2.24 at T1; 14.32, DS 2.53 at T2. MMSE control group: 14.25, DS 2.66 at T0; 14.24, DS 2.88 at T1; 14.00, DS 2.97 at T2.
Li et al., 2016 [50]	Randomized study	81 patients divided into two groups (46 males). CASES (citicoline sodium capsules orally 200 mg three times a day): 41 patients. CONTROLS (basic medications such L-dopa or pramipexole with the matching placebo): 40 patients.	Mild cognitive impairment in Parkinson’s disease	Primary outcome: the effect of citicoline adjuvant therapy on mild cognitive impairment in Parkinson’s disease using MoCA and SCOPA-COG evaluations. MoCA and SCOPA-COG evaluations were performed at baseline, 12 and 18 months.	MoCA scale scores (baseline): cases = 24.03, DS 3.22-controls = 23.89, DS 2.27. MoCA scale scores (after 12 months): cases = 23.65, DS 2.55-controls = 22.53, DS 4.14. MoCA scale scores (after 18 months): cases = 23.12, DS 2.8-controls = 21.49, DS 3.99. SCOPA-COG scale scores (baseline): cases = 23.79, DS 2.82-controls = 23.43, DS 2.19. SCOPA-COG scale scores (after 12 months): cases = 21.55, DS 3.05-controls = 20.73, DS 4.14. SCOPA-COG scale scores (after 18 months): cases = 21.09, DS 2.78-controls = 19.25, DS 3.68.

AD: Alzheimer’s disease; MD: mixed dementia; MCI: Mild Cognitive Impairment, AChEI: acetylcholinesterase inhibitors; MMSE: Mini Mental State Examination; MMSEc: Mini Mental State Examination corrected according to age and education, GCI: global cognitive impairment; EuroQoL-5D: euro quality of life—5 dimensions; MoCA: Montreal Cognitive Assessment; SCOPA-COG: SCales for Outcomes in PArkinson’s disease-COGnition; TRT: Total Reaction Time; RRT: Recognition Reaction Time; MRT: Movement Reaction Time; CFFA: Critical Fusion Frequency; CFFD: Critical Flicker Frequency; WM-1: Working Memory-one back; WM-2: Working Memory-two back; WM-3: Working Memory-three back; PT: processing time; RCT: Randomized Controlled Trial; DS: standard deviation.

**Table 3 nutrients-15-00386-t003:** Risk of bias of the included studies.

	Random Sequence	Allocation Concealment	Performance Bias	Detection Bias	Attrition Bias	Reporting Bias	Competing Risk	Overall Risk of Bias
Alvarez-Sabin et al., 2016 [46]	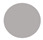						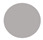	
Castagna et al., 2016 [44]							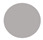	
Castagna et al., 2021 [45]							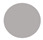	
Gareri et al., 2016 [48]							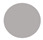	
Cotroneo et al., 2013 [47]	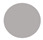				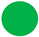		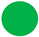	
Li et al., 2016 [50]	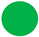	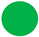			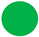	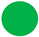		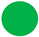
Castagna et al., 2021 [49]							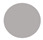	

(RED = high risk of bias; YELLOW = unclear risk of bias; GREEN = low risk of bias; GREY = unreported)

## 3. Results

We included seven studies, of which only two were randomized clinical trials. Four studies [44,45,48,49] enrolled patients with Alzheimer’s Disease, two enrolled patients with cognitive impairment (one [50] enrolled patients with MCI associated with Parkinson’s disease and the other enrolled patients with mild vascular cognitive impairment [47]), and one enrolled patients with post-stroke dementia [46]. Three studies focused on prevention of cognitive decline or dementia in people with MCI [46,47,50] and four evaluated the use of citicoline in the treatment of dementia [44,45,48,49].

Citicoline was either used as a single intervention [46,47,50] or in addition to standard therapy [44,45,48,49]; the control group was either composed of placebo [46,47,50] or standard therapy [44,45,48,49]. The dose regimen and route of administration of citicoline varied among the studies and different cognitive scales were used to assess the impact of citicoline on cognitive functions; the most used being the Mini-mental State Examination (MMSE) (Table 2). The sample sizes were on average relatively small (213 patients), and the follow-up times varied from nine months to two years.

All the studies showed a positive effect of citicoline on cognitive function. Cotroneo et al. [47] showed that in patients with mild vascular cognitive impairment the MMSE score over 9 months remained stable in the group receiving citicoline, while it declined in the control group (−1.9 points between baseline and follow-up). Similar results were reported by Castagna et al. in three recent studies [44,45,49]: patients with AD or mixed dementia treated with citicoline had higher MMSE scores compared with the control group, in which the MMSE score showed a significant decrease. The same results were reported in the study by Gareri et al. [48] in patients with AD. These results were also consistent in a different population (post-stroke patients) [46], in which the group receiving citicoline had a significant reduction of cognitive impairment prevalence during follow up (GCI 43.6% at 1 month, 32.5% at 6 months, 29% at 1 year, 27.9% at 2 years) and lower prevalence of cognitive impairment at 2 years compared with controls (27.9% vs. 39% in the control group). Li et al. [50] evaluated patients suffering from MCI associated with Parkinson’s disease and analyzed the effects of citicoline on cognitive functions, assessed by MOCA (Montreal Cognitive Assessment) and SCOPA (SCales for Outcomes in PArkinson’s disease-COGnition), and found a significant difference between the treatment and control groups after 12 (cases = 21.55, DS 3.05; controls = 20.73, DS 4.14; *p* < 0.05) and 18 (cases = 21.09, DS 2.78; controls = 19.25, DS 3.68; *p* < 0.01) months of treatment.

One of the studies [46] could not be included in the meta-analysis of data because it reported the proportion of patients with improvement rather than the actual difference in the scores. When we pooled the results of the remaining six studies (one randomized trial), we found sizeable differences according to the different scenarios made for calculating standard deviations. The most conservative estimate of the standardized mean difference in the outcome measures was 0.56 (95% CI: 0.40–0.72), and the results were virtually unchanged when only observational studies were included (0.56, 95% CI 0.37–0.75). We also estimated the pooled mean difference of MMSE score in the five studies that reported this outcome measure, finding a difference of 1.55 (95% 0.74–2.37) (Figure 2). In the most optimistic scenario, we estimated an overall standardized mean difference of the outcome measures of 0.91 (95% CI: 0.49–1.34), but in the studies reporting the MMSE score, the results were identical (1.57, 95% CI: 0.77–2.37) (Figure 3).

All the funnel plots (Appendix A) were very asymmetrical, indicating the presence of bias. Actually, the risk of bias was on average high in all the studies, especially with respect to selection bias (random sequence and allocation concealment), performance bias and detection bias (Table 3). In addition, competing risk cannot be assessed since the data are not reported in the articles.

The quality of the evidence evaluated using the GRADE framework was very low, except for one study [50] which was moderate (Table 4).

## 4. Discussion

All the studies included in this systematic review indicate that citicoline is effective in preventing or slowing down cognitive decline and has a favorable safety profile [35,51]. The estimates of the effect were consistent across the sensitivity analyses performed using different calculations of standard deviations and including only studies with the same design. Consistent with the experimental evidence that the effects of citicoline are more evident with long-term administration, all of the studies showed that the effect size tended to increase over time.

These positive results should be interpreted also in light of the fact that all the studies had an intermediate/high risk of bias, and the overall quality of the evidence was low. With regard to safety, as most studies revealed an unclear risk for recall/reporting bias, adverse events may have been underreported. Furthermore, rarer events may not have been observed because of limited sample sizes and follow-up duration.

Our results confirm the evidence on the positive effects of citicoline in a wide range of neurological conditions, such as dementia, neuropathic pain and nerve regeneration [39]. They are in line with and extend those of Piamonte et al. [37], which showed that citicoline in addition to standard treatment has beneficial effects on cognition in people with AD. Nonetheless, it is difficult to make a comparison between our results and those of this review because it only included two observational studies in patients with AD in which citicoline was used as an adjuvant in addition to the standard treatment with ACheI. We also expand the evidence provided by Que et al. [38], who showed that in patients with Parkinson’s disease citicoline as an adjuvant therapy has positive effects on delaying the progression of cognitive impairment. However, a major limitation of that review is the significant heterogeneity of the studies in terms of design and measured outcomes that prevented the pooling of results; moreover, the included studies were dated 1991 or earlier. Our promising results, however, must be evaluated against the poor quality of the available evidence, with the inherent risk of bias in favor of the intervention. Furthermore, the observed effect in our review is an increase in MMSE score of 1.5 point in patients treated with citicoline, but considering the existing literature [52], the minimal clinically important difference (MCID) estimated for MMSE is -2 for mild-AD and -3 for moderate-severe AD over 1 year (where “-” indicates a decrease in score). However, the only study on MCI [47] shows an increase in MMSE score of 2.4 points in patients treated with citicoline, which is considered clinically significant according to the evidence [52].

## 5. Conclusions

Our results confirm the positive effects of citicoline shown in other studies, but at the same time, they show that the available evidence is of poor quality and likely to be biased. Thus, well designed, high-quality randomized clinical trials are needed to qualify citicoline use for the prevention or treatment of cognitive decline. Meanwhile, citicoline should be used according to clinical judgement and taking into account the limitations of the available evidence.

## Figures and Tables

**Figure 1 nutrients-15-00386-f001:**
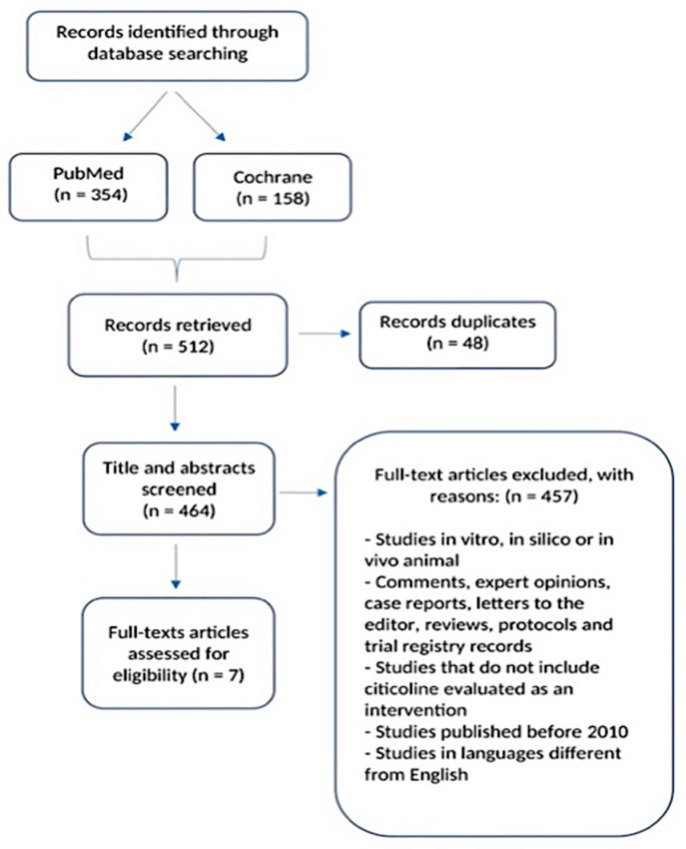
PRISMA flow diagram illustrating the study selection process [41].

**Figure 2 nutrients-15-00386-f002:**
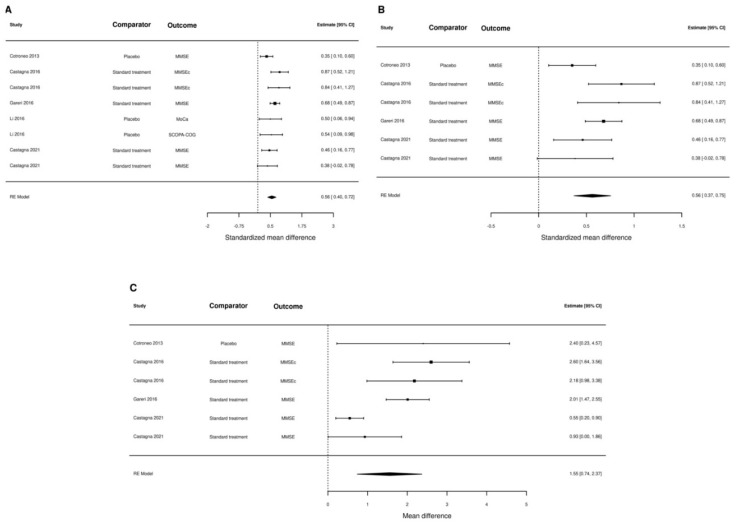
Forest plot summarizing all studies (panel (**A**)), only observational studies (panel (**B**)), and only studies using MMSE as outcome measure (panel (**C**)). SD were calculated assuming a small correlation coefficient (r = 0.6).

**Figure 3 nutrients-15-00386-f003:**
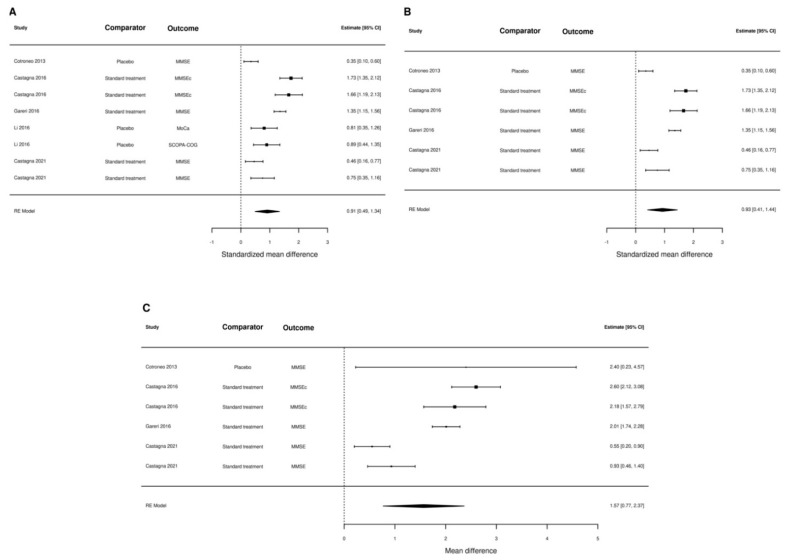
Forest plot summarizing all studies (panel (**A**)), only observational studies (panel (**B**)), and only studies using MMSE as outcome measure (panel (**C**)). SD were calculated assuming a large correlation coefficient (r = 0.9).

**Table 1 nutrients-15-00386-t001:** Inclusion criteria based on PICO algorithm.

Patient (P)	Adults with normal cognition, MCI, Alzheimer’s disease or vascular dementia. There were no restrictions on sex, ethnicity or severity of the cognitive impairment at baseline.
Intervention (I)	Citicoline as dietary supplements. Co-interventions with citicoline and standard treatment were allowed.
Comparison (C)	Standard of care, no intervention or placebo.
Outcome (O)	Incidence of all-cause dementia or mild cognitive impairment; Cognitive functions measured by cognitive scales (MMSE, MocA and SCOPA-cog), global cognitive impairment (GCI), human vigilance and visual working memory.
Type of studies (S)	Both randomized controlled clinical trials (RCTs) and clinical studies were included; Articles published from 2010 to September 2022 in English.

MCI: Mild Cognitive Impairment; MMSE: Mini Mental State Examination; MoCA: Montreal Cognitive Assessment; SCOPA-cog: SCales for Outcomes in PArkinson’s disease-COGnition; GCI (global cognitive impairment); RCT: Randomized Controlled Trial.

**Table 4 nutrients-15-00386-t004:** GRADE framework (grading of recommendations, assessment and evaluation).

Scheme.	A Priori Ranking	Upgrade/Downgrade	Final Grade	Factors Affecting Recommendation	Make Recommendation
Castagna 2016 [44]	LOW	Downgrade	VERY LOW	Low side effects	Weak for using
Castagna 2021 [49]	LOW	Downgrade	VERY LOW	Low side effects	Weak for using
Castagna 2021 [45]	LOW	Downgrade	VERY LOW	Low side effects	Weak for using
Gareri 2016 [48]	LOW	Downgrade	VERY LOW	Low side effects	Weak for using
Alvarez-Sabin 2016 [46]	LOW	Downgrade	VERY LOW	Low side effects	Weak for using
Cotroneo 2013 [47]	LOW	Downgrade	VERY LOW	Low side effects	Weak for using
Li 2016 [50]	HIGH	Downgrade	MODERATE	Low side effects	Strong for using

## Data Availability

Not applicable.

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
