# Peer review of "Is Citicoline Effective in Preventing and Slowing Down Dementia?—A Systematic Review and a Meta-Analysis"

_nutrients, 2023, doi:10.3390/nu15020386_

Round 1

Reviewer 1 Report

The authors of the manuscript performed a systematic review and meta-analysis to assess whether citicoline, compared to placebo or standard treatment, is effective in preventing dementia in patients with mild cognitive impairment or in improving cognitive function in persons with dementia. 

After a rigorous review of 464 articles by two of the authors, only 8 articles that met the inclusion criteria were included in this systematic review. At the end of the systematic review strategy, although the authors found positive effects of citicoline therapy from the published data, they clearly emphasise the need to improve the quality of the studies by pointing out that the available evidence is not scientifically rigorous. The authors suggest the design of adequate and rigorous randomised clinical trials to define the therapeutic appropriateness of citicoline for the prevention or treatment of cognitive decline. In conclusion, it is considered that the careful analysis presented in this manuscript , will open a broad scientific discussion on the real therapeutic benefit of citicoline in patients with cognitive decline.

Author Response

 We thank the reviewer for the positive comments on our work. 

Reviewer 2 Report

Dear Authors,

first of all, I wish to thank You for giving me the opportunity to read this Your manuscript submitted for publication in Nutrients. 

Here are my comments and suggestions. I hope they will help to improve scientific soundness and quality of presentation of Your review article. 

MAJOR COMMENTS

I have three major comments.

1)  You chose to include the Al-kuraishy and Al-Gareeb's article (reference no. 42) in Your review. I cannot agree with this decision because it is not in line with the topic You want to explore. As You know, these researchers determined the effect of citicoline on healthy volunteers having an age range of 21-22 years. These healthy and young volunteeers had a very short follow-up (only two weeks). Therefore, including data from this article seems to me a serious inclusion bias.  I suggest that You remove this article and modify the whole text accordingly.   

2)  Your Table 3 (Risk of bias of the included studies) should be revised. You attributed different colors to articles written by the same research group (Castagna, Gareri, Cotroneo). These researchers had identical (or very similar) methodology. Can You please explain in detail Your choices? 

3)  You concluded that "the available evidence lacks quality and is likely to be biased"  and that "well designed, high-quality randomized clinical trials are needed to quality citicolice use for the prevention or treatment of cognitive decline" (lines 283-285). This seems a generic statement, not useful to the readers. Please, can You write something constructive and purposeful? Our readers will appreciate it very much.        

MINOR COMMENTS

1.  Tables no. 2, 3 and  4: ALVAREZ-SABIN (and not SABIN) et al.   

2.  One of the articles included in Your systematic review concerns the effect of citicoline adjuvant therapy on mild cognitive impaiment in Parkinson's disease (Li et al, reference no. 39). I suggest You write in the Introduction section something on MCI/dementia and Parkinson's disease, how did You write about Alzheimer' s diseases and vascular dementia (lines from 43 to 58).    

3.  References should be extended.  

4.  Some typo errors have to be corrected.  

Author Response

MAJOR COMMENTS

Point 1: You chose to include the Al-kuraishy and Al-Gareeb's article (reference no. 42) in Your review. I cannot agree with this decision because it is not in line with the topic You want to explore. As You know, these researchers determined the effect of citicoline on healthy volunteers having an age range of 21-22 years. These healthy and young volunteeers had a very short follow-up (only two weeks). Therefore, including data from this article seems to me a serious inclusion bias.  I suggest that You remove this article and modify the whole text accordingly.   

Response 1: We thank the reviewer for this comment. The article was included because it met our inclusion/exclusion criteria (clinical trials or randomized controlled trials; presence of control group; quantitative outcomes; publication date (from 2010 to September 2022); studies performed on humans; studies in English language). Nonetheless, we see the point of the reviewer and we have modified the paper as suggested.

Point 2:   Your Table 3 (Risk of bias of the included studies) should be revised. You attributed different colors to articles written by the same research group (Castagna, Gareri, Cotroneo). These researchers had identical (or very similar) methodology. Can You please explain in detail Your choices? 

Response 2: We based our judgement on what the authors reported in the Method section of their articles. For example, Castagna 2021 and Cotroneo 2013 were open label studies, so they have a high risk of performance and detection bias, while in Gareri 2016 and Castagna 2016 it was specified that it was a retrospective case-control study, so they have an unclear risk of these bias. Likewise, Cotroneo is the only one with a low risk of attrition bias because it reported that less than 10% of patients dropped out during the follow up. Moreover, Cotroneo is the only one with a low competing risk because it reported that only two patients died during the follow up, while all the other studies do not report this information.

Point 3: You concluded that "the available evidence lacks quality and is likely to be biased" and that "well designed, high-quality randomized clinical trials are needed to quality citicolice use for the prevention or treatment of cognitive decline" (lines 283-285). This seems a generic statement, not useful to the readers. Please, can You write something constructive and purposeful? Our readers will appreciate it very much. 

Response 3: Our conclusion was based on the assumption that only well designed and conducted controlled clinical trials are able to inform on the effectiveness of an intervention. We tried to make our conclusion more useful by underlining that lacking this information “citicoline should be used according to clinical judgement and taking into account the limitations of the available evidence”.

Conclusion (revised version): “
Our results confirm the positive effects of citicoline shown in other studies, but at the same time show that the available evidence is of poor quality and likely to be biased. Thus, well designed, high-quality randomized clinical trials are needed to qualify citicoline use for the prevention or treatment of cognitive decline.  Meanwhile, citicoline should be used according to clinical judgement and taking into account the limitations of the available evidence.”

  MINOR COMMENTS

Point 1: Tables no. 2, 3 and 4: ALVAREZ-SABIN (and not SABIN) et al. 

Response 1: We thank the reviewer for pointing out this mistake, we corrected the name.

Point 2: One of the articles included in Your systematic review concerns the effect of citicoline adjuvant therapy on mild cognitive impaiment in Parkinson's disease (Li et al, reference no. 39). I suggest You write in the Introduction section something on MCI/dementia and Parkinson's disease, how did You write about Alzheimer' s diseases and vascular dementia (lines from 43 to 58). 

Response 2: Thank you for this comment, we added the following text to the Introduction.

Introduction (revised version): “..
Almost half of stroke survivors have cognitive impairment[12]; cognitive decline after stroke is even more common than stroke recurrence, and stroke patients with cognitive impairment but no dementia have an increased risk of developing VaD and other dementias at 5 years [13]. MCI and dementia can also occur in patients with Parkinson disease (PD); the prevalence of PD-MCI ranges from 20% to 40% depending on the population studied and the incidence of PD-dementia increases with duration of disease, with estimates as high as up to30% at 5 years and over 80% at 20 years from diagnosis[14,15].  Cognitive decline is usually slow and insidious, but may be rapid in some cases[16]. MCI and dementia pose significant challenges to patients, caregivers and healthcare systems due to their rising prevalence and socioeconomic burden, with numbers of people living with dementia worldwide projected to triple to 152 million cases by 2050..”

Point 3: References should be extended.

Response 3: We added references on dementia in Parkinson’s disease and added extra references in the Introduction.

Point 4:  Some typo errors must be corrected.  

Response 4: We apologize for the mistakes; the English language has been extensively revised.

Round 2

Reviewer 2 Report

Dear Authors,

I read the revised version of Your manuscript, and I found that all my comments and suggestions were satisfactorily met.

Thanks !

My overall recommendation is that this Your review article can be published in its present form. 

Best regards.